# Early warning scores for detecting deterioration in adult hospital patients: a systematic review protocol

Stephen Gerry,[1] Jacqueline Birks,[1] Timothy Bonnici,[2] Peter J Watkinson,[3] Shona Kirtley,[4] Gary S Collins[1]

[1]Centre for Statistics in Medicine, Nuffield Department of Orthopaedics, Rheumatology & Musculoskeletal Sciences, University of Oxford, Oxford, UK
[2]Nuffield Department of Medicine, University of Oxford, Oxford, UK
[3]Kadoorie Centre for Critical Care Research and Education, Nuffield Department of Clinical Neurosciences, University of Oxford, Oxford, UK
[4]UK EQUATOR Centre, Centre for Statistics in Medicine, Nuffield Department of Orthopaedics, Rheumatology & Musculoskeletal Sciences, University of Oxford, Oxford, UK

Correspondence to
Stephen Gerry;
stephen.gerry@csm.ox.ac.uk

## ABSTRACT

**Introduction** Early warning scores (EWSs) are used extensively to identify patients at risk of deterioration in hospital. Previous systematic reviews suggest that studies which develop EWSs suffer methodological shortcomings and consequently may fail to perform well. The reviews have also identified that few validation studies exist to test whether the scores work in other settings. We will aim to systematically review papers describing the development or validation of EWSs, focusing on methodology, generalisability and reporting.

**Methods** We will identify studies that describe the development or validation of EWSs for adult hospital inpatients. Each study will be assessed for risk of bias using the Prediction model Risk of Bias ASsessment Tool (PROBAST). Two reviewers will independently extract information. A narrative synthesis and descriptive statistics will be used to answer the main aims of the study which are to assess and critically appraise the methodological quality of the EWS, to describe the predictors included in the EWSs and to describe the reported performance of EWSs in external validation.

**Ethics and dissemination** This systematic review will only investigate published studies and therefore will not directly involve patient data. The review will help to establish whether EWSs are fit for purpose and make recommendations to improve the quality of future research in this area.

**PROSPERO registration number** CRD42017053324.

## BACKGROUND

Towards the end of the 20th century, accumulating evidence suggested that people in hospital wards were dying and suffering harm unnecessarily.[1–3] Multiple studies have demonstrated that cardiac arrest or death is commonly preceded by several hours of deranged physiology.[4–6] Recommendations were made to put systems in place to use this information to identify and respond to previously unrecognised deterioration in patients.[7] In response, the first early warning score (EWS) was published in 1997.[8]

EWSs are simple tools to reduce unnecessary harm in hospitals. These clinical

## Strengths and limitations of this study

► The first systematic review in a decade to include all published early warning scores (EWSs).
► The first systematic review to include EWS validation studies.
► The review will assess the methodology and generalisability of studies to identify the best current EWSs and make recommendations for future development and validation studies.
► The review will be limited to examining published EWSs. Many other scores may be in clinical use, but not published.

prediction models use patients' measured vital signs to monitor their health during their hospital stay and identify their likelihood of deteriorating, characterised as death or admission to intensive care unit (ICU), for example. Should a patient show signs of deteriorating, the EWS triggers a warning so that care can be escalated. EWSs, which are also commonly referred to as track-and-trigger scores, are often implemented as part of an 'early warning system' or 'EWS system'. These are computer systems which record vital signs, automatically or manually and then implement the EWS algorithm to indicate a patient's risk of deterioration. The interest of this review lies in the underlying scoring systems/algorithms themselves and not the systems in which they are implemented.

There are now many EWSs available.[9–11] They are routinely used in several countries, including the Netherlands, USA and Australia and their use in UK hospitals is mandated as a standard of care by the National Institute For Health and Clinical Excellence (NICE).[12] Based on the Hospital Episode Statistics,[13] we estimate that EWSs are used more than 120 million times per year in the NHS in England alone, a

conservative estimate that probably well underestimates the true total.[i]

EWSs have been derived using a variety of approaches. Some have been developed using statistical methods for clinical prediction, by linking observations (eg, vital signs) to outcomes (eg, death, ICU admission) through regression models. Others have been based on clinical consensus without statistical modelling. Although there is now an abundance of clinical prediction models in many fields of medicine and healthcare, in practice many of these models are scarcely used.[14 15] Systematic reviews of clinical prediction models in other clinical areas have all concluded that many are poorly developed[15–17] and that they are rarely and inappropriately evaluated[18 19] (often referred to as validation), that is, tested in different settings to which they were developed. There is no common agreement on which of the dozens of EWSs available performs best. Most problematically, recent evidence suggests that EWSs have not solved the problem they were designed for: unrecognised deterioration of patients in hospitals remains a major issue.[20]

The aim of this systematic review is to critically appraise papers describing the development and validation of EWSs for adult hospital inpatients, with a particular focus on methodology, reporting and generalisability, in order to identify high quality EWSs and provide guidance regarding the methods to develop and validate future EWSs.

### Existing systematic reviews

Four systematic reviews of studies which develop or validate EWSs have been published.[9–11] Those by Gao et al[9] and GB Smith et al[10] were published almost a decade ago, while MEB Smith et al[11] used narrow inclusion criteria and did not include all available EWSs,[11] and the review by Kyriacos et al[21] was a more general overview of the literature. Several new EWSs have been published since.

The main aims of the reviews were to describe the development of EWSs, assess their predictive performance and assess any impact studies that evaluate the effect of implementing EWSs in clinical practice. Other reviews, such as those by Alam et al[22] and McGaughey et al,[23] looked at impact studies, but we do not plan to include these in our review.

Many of the reviewed scores included similar predictors and applied similar weights to those predictors. Nearly all of the scores included pulse rate, breathing rate, systolic blood pressure and temperature. The reviews also found some indication that scores that included age performed better.[10] In contrast to studies developing EWSs, validation studies that evaluated the performance of EWSs were relatively uncommon.

---

[i](~12 million non-day-cases per year * mean length of stay 5 days * 2 observations per day).

The use of poor methods to develop EWSs could mean that the scores are unreliable and fail to accurately predict risk. Gao et al[9] and MEB Smith et al[11] subjectively reported that they found many of the primary studies to be of low quality, used suboptimal methods and were at high risk of bias.[9 11] However, none of the reviews made a detailed and structured evaluation of the approaches used to develop EWSs, following recommended methodological considerations in the field of clinical prediction models.[24–28]

After a prediction model (ie, an EWS) has been developed, its predictive accuracy should be evaluated in the same population used to derive it, a process called internal validation. The two widely recommended characteristics that describe the performance of a prediction model are discrimination (eg, the c-index and AUROC) and calibration.[24] Discrimination reflects a prediction model's ability to differentiate between those who develop an outcome (ie, death) and those who do not. A model should predict higher risks for those who develop the outcome. Calibration reflects the level of agreement between observed outcomes and the model's predictions.

Both discrimination and calibration must be assessed and reported to judge a model's accuracy.[24] However, as in many other clinical areas, studies evaluating EWSs have tended to give more prominence to discrimination and have rarely assessed model calibration. Two of the reviews investigated how primary EWS studies report predictive performance, with conflicting conclusions. Gao et al[9] found unacceptance predictive performance,[9] whereas MEB Smith et al[11] found good predictive performance. This difference in result may reflect differences in the included studies and how the authors assessed model performance.

Internal validation provides insights into model performance in the same population used to derive the model. In contrast, external validation assesses the model's performance in a different population from that used to derive it. External validation assesses model discrimination and calibration to determine whether the model performs satisfactorily in data other than that it was developed with, which is called generalisability.[29] Although the four reviews did not have a specific focus on external validation studies, they all highlighted a lack of external validation studies of EWSs. GB Smith et al[10] did not investigate validation studies, but performed their own external validation as part of their review by evaluating the identified models using their own data. They found that none of the scores showed good enough performance.[10]

### Research aims

In this systematic review, we aim to identify all existing published EWSs for adult hospital inpatients and:
1. Describe and critically appraise the methods that have been used to develop and validate (where appropriate) the scores. We will take a wide-ranging approach and will cover statistical aspects, such as how missing data are accounted for and how continuous

predictors are used. We will also investigate aspects of generalisability, such as details of the populations used to develop the models.
2. Describe which predictors are included in the scores and how they are weighted.
3. Report which EWSs have undergone external validation and, if so, how well they performed.

## METHODS

Our systematic review protocol was registered with the International Prospective Register of Systematic Reviews (PROSPERO) on 12 July 2017 (registration number CRD42017053324). Our systematic review will be carried out and reported in accordance with two published guidelines: the Critical Appraisal and Data Extraction for Systematic Reviews of Prediction Modelling Studies (CHARMS) checklist[30] and the Preferred Reporting Items for Systematic Reviews and Meta-Analyses (PRISMA) checklist.[31]

### Selection criteria

We will include studies that satisfy all of the following criteria:
1. The study describes the development or validation of one or more EWSs, defined as a score used to identify hospitalised patients at risk of clinical deterioration.
2. The EWS studied combines information from at least two predictor variables to produce a summary risk estimate.
3. Validation studies will only be included where the corresponding development articles are available.
   We will exclude papers where any of the following apply:
1. The score was developed for use in a subset of patients with a specific disease or group of diseases.
2. The score was developed for use with children (aged under 16 years) or pregnant women.
3. The score is intended for outpatient use.
4. The score is intended for use in the ICU.
5. Reviews, letters, personal correspondence and abstracts.

### Search strategy

Studies will be identified by searching the medical literature using Medline (OVID), CINAHL (EbscoHost) and Embase (OVID) to identify primary articles reporting on the development and/or validation of EWSs. We will use a combination of relevant controlled vocabulary terms for each database (eg, MeSH, Emtree) and free-text search terms. No date or language restrictions will be applied. Citation lists of previous systematic reviews and included studies will be searched to identify any studies missed by the search. We will also conduct a Google Scholar search to identify any other eligible studies. Online supplementary appendix A shows a draft search strategy.

### Study selection

Two reviewers will independently screen all titles and abstracts using prespecified screening criteria. The full text of any relevant articles will then be independently assessed by two reviewers. Disagreements will be resolved by discussion and, if necessary, referral to a third reviewer. The study selection process will be reported using a PRISMA flow diagram.[31]

### Data extraction

Data will be independently extracted by two reviewers using a standardised and piloted data extraction form. The form will be administered using the Research Electronic Data Capture (REDCap) electronic data capture tool.[32] Disagreements will be resolved by discussion and, if necessary, by referral to a third reviewer. We will choose items for extraction based on the CHARMS checklist,[30] supplemented by subject-specific questions and methodological guidance. Items for extraction will include:
► Study characteristics (development and validation) (eg, country, year).
► Study design (development and validation) (eg, prospective, case control, cohort, clinical consensus).
► Patient characteristics (development and validation) (eg, hospital ward, age, sex).
► Predicted outcome (development and validation) (eg, survival at 24 hours, ICU admission at 24 hours).
► Model development (development) (eg, sample size, type of model, handling of continuous variables, selection of variables, missing data, method of internal validation).
► Model presentation (development) (eg, full regression model, simplified model, risk groups).
► Assessment of performance (development and validation) (eg, measures of discrimination, measures of calibration).

### Assessment of bias

Each article will be independently assessed by two reviewers using the Prediction model Risk of Bias ASsessment Tool (PROBAST), which was recently developed by the Cochrane Prognosis Methods Group to assess the quality and risk of bias for prediction models (due to be submitted shortly; Wolff R, Whiting, Mallett S *et al.* (*including author GSC*), personal communication). PROBAST consists of 23 signalling questions within four domains (participant selection, predictors, outcome and analysis).

### Evidence synthesis

We will summarise the results using descriptive statistics, graphical plots and a narrative synthesis. We do not plan to perform a quantitative synthesis of the scores or their predictive performance. However, if we identify multiple studies that evaluate the same EWS and report common performance measures, we will summarise their performance using a random-effects meta-analysis.[33] The PROBAST evaluation will be used to determine the models' risk of bias, including whether the EWSs are likely to work as intended for the hospital population of interest. The models will be classed as low, high or unclear risk of bias.

## DISCUSSION

Although EWSs are extensively used in clinical practice, the methodology behind them remains questionable. Although not formally assessed, previous systematic reviews of EWSs have indicated that many studies suffer from a lack of quality and that few EWSs have been satisfactorily validated.[9–11] These aspects are crucial for developing a prediction model that can confidently be rolled out into clinical practice. This systematic review will bridge this important gap by examining methodological quality and external validation in detail. This systematic review is timely, as it is now nearly a decade since the last comprehensive review of EWSs, which have only existed for 20 years.

EWSs have historically been implemented as part of traditional paper observation charts. The requirement for scores to be calculated manually necessitated the use of simple scoring algorithms. Storage of data on paper has been a barrier to collection of large datasets for score derivation and validation. Digital systems are increasingly being used to record vital signs and calculate EWSs,[34] offering the opportunity to be more rigorous and innovative in the development and implementation of new EWSs. The adoption of digital vital signs charting offers an opportunity to transition away from poor quality EWSs. Our review will provide the evidence for creators of digital systems to identify which EWSs should be prioritised for implementation.

**Contributors** SG, JB, TB, PJW and GSC conceived the study. SG developed the study protocol and will implement the systematic review under the supervision of GSC. SG will provide the study's statistical analysis plan and will analyse the data. SG and SK will perform the study search and SG will screen and extract the data. JB, TB, PJW and GSC will review the work. SG wrote the first protocol manuscript draft and all authors gave input into and approved the final draft of the protocol.

**Funding** SG is funded by an NIHR Doctoral Fellowship (DRF-2016-09-073). JB, TB, PJW and GSC are supported by the NIHR Biomedical Research Centre, Oxford. The funders have not played any role in the development of this protocol.

**Competing interests** None declared.

**Provenance and peer review** Not commissioned; externally peer reviewed.

**Data sharing statement** All unpublished data will be made available on request. The first author, SG, should be contacted with requests for unpublished data.

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
