## [Reviewer comments · BMJ Open]

ARTICLE DETAILS

TITLE (PROVISIONAL)	Early Warning Scores for detecting deterioration in adult hospital patients: a systematic review protocol
AUTHORS	Gerry, Stephen; Birks, Jacqueline; Bonnici, Tim; Watkinson, Peter; Kirtley, Shona; Collins, Gary

VERSION 1 – REVIEW

REVIEWER	Damian Roland Leicester University and Hospitals Leicester UK Co-investigator on PUMA study (NIHR study looking at PEWS £1.8 million) Lead investigator on PAT-POPS study (NIHR Study look at ED scoring systems £300000)
REVIEW RETURNED	05-Sep-2017

GENERAL COMMENTS	Shouldn't Early Warning System be a keyword as well as Early Warning Score?
---

REVIEWER	Mary McLellan Boston Children's Hospital United States
REVIEW RETURNED	06-Sep-2017

GENERAL COMMENTS	This is a timely systematic review as the number of EWSs has been exponentially increasing. The authors indicated they would be excluding EWSs used in an outpatient setting. If the authors are also excluding EWSs used in ICUs they should indicate that in the exclusion criteria. I look forward to your results!
--

REVIEWER	Veronica Lambert Associate Professor Dublin City University Ireland
REVIEW RETURNED	11-Sep-2017

GENERAL COMMENTS	Many thanks for inviting me to review this paper on a issue that is topical with the increasing development and use of early warning scoring systems for detecting deterioration in hospital patients. After reading the manuscript I had some minor thoughts to consider: In the abstract I was a little unclear on the use of the heading ethics and dissemination. I think it would be useful to provide an operational definition for EWS for this protocol considering the use of many terms in the literature such as early warning scores, tools, systems etc. I am gathering from the protocol that the specific focus is on the scoring tool as opposed to a system perspective which is often where the complexity of the early warning detection and response lies; how will this be taken into account in this review or is this beyond this review which seeks to merely look at scoring tool validation? In a similar vain define validation at the outset especially for readers not familiar with this concept. There is some reference to this within the existing SR section of the paper but I wondered does it sit there or should there be two separate sections here one critiquing previous reviews and gaps therefore building rationale for the need for this particular review and what it will add to the knowledge base on EWS and a separate section that defines and outlines what validation is and as such operationally defines that for the context of this review. Three existing systematic reviews are referred to at the outset of the protocol why were these three specifically selected as there are numerous other published reviews on EWS (e.g. a quick search revealed O'Neill 2013, Alam 2014, Kynacos 2011, McGaughey 2010, Saab 2017) - are these relevant to refer to acknowledge and then to situate this proposed review within that body of evidence in terms of what it will contribute in terms of gaps in systematic reviews on EWS. I do think this review attempts to address a gap by focusing specifically on tool validation however the arguments are currently a little weak on this. It is not clear from the outset of the paper (including in the abstract) that the protocol for EWS review is specific to adult inpatients in hospital and this would be good to clarify from the beginning. In relation to the review selection criteria it might be useful to consider structuring it according to PICOS or referring to these.
--

VERSION 1 – AUTHOR RESPONSE

Reviewer: 1

Comment: Shouldn't Early Warning System be a keyword as well as Early Warning Score?

Response: Thank you. We have now included early warning system in the list of keywords.

Reviewer: 2

Comment: This is a timely systematic review as the number of EWSs has been exponentially increasing. The authors indicated they would be excluding EWSs used in an outpatient setting. If the authors are also excluding EWSs used in ICUs they should indicate that in the exclusion criteria. I look forward to your results!

Response: Many thanks for your positive feedback. You are correct that we will be excluding ICU specific scores, and have added a sentence in the 'selection criteria' section to clarify this.

Reviewer: 3

Comment: Many thanks for inviting me to review this paper on a issue that is topical with the increasing development and use of early warning scoring systems for detecting deterioration in hospital patients. After reading the manuscript I had some minor thoughts to consider: In the abstract I was a little unclear on the use of the heading ethics and dissemination.

Response: Many thanks for your helpful comments on the manuscript. Regarding the ethics and dissemination section of the abstract, we have left this in according to the journal policy.

Comment: I think it would be useful to provide an operational definition for EWS for this protocol considering the use of many terms in the literature such as early warning scores, tools, systems etc. I am gathering from the protocol that the specific focus is on the scoring tool as opposed to a system perspective which is often where the complexity of the early warning detection and response lies; how will this be taken into account in this review or is this beyond this review which seeks to merely look at scoring tool validation?

Response: Thank you, this is very helpful. We have included extra information in the background section to clarify the relationship between 'early warning scores' and 'early warning systems'. We have also clarified that it is the scores (or algorithms) themselves that are the main interest of this paper.

Comment: In a similar vain define validation at the outset especially for readers not familiar with this concept. There is some reference to this within the existing SR section of the paper but I wondered does it sit there or should there be two separate sections here one critiquing previous reviews and gaps therefore building rationale for the need for this particular review and what it will add to the knowledge base on EWS and a separate section that defines and outlines what validation is and as such operationally defines that for the context of this review.

Response: We have added a further definition of validation into the introduction section. We purposefully chose not to include too many definitions, so as to not overload the paper. But as you say this one is important to the paper.

Comment: Three existing systematic reviews are referred to at the outset of the protocol why were these three specifically selected as there are numerous other published reviews on EWS (e.g. a quick search revealed O'Neill 2013, Alam 2014, Kynacos 2011, McGaughey 2010, Saab 2017) - are these relevant to refer to acknowledge and then to situate this proposed review within that body of evidence in terms of what it will contribute in terms of gaps in systematic reviews on EWS. I do think this review attempts to address a gap by focusing specifically on tool validation however the arguments are currently a little weak on this.

Response: You are correct – the main aim of our review is to look at the methodology, reporting and generalisability of studies which report either the development or validation of early warning scores. The three reviews referred to in the 'existing systematic reviews' section were the ones which dealt with these ideas. The review by Kyriacos probably also deserves to be included in this section as well, which has been amended in the new submission, along with some clarification of why we specifically refer to these four reviews. The reviews by Alam and McGaughey concentrate primarily on the effect of implementing early warning scores/early warning systems, and so are not directly relevant to our review. We previously mentioned Alam's review briefly, but have modified this in the text to become more inclusive.

Comment: It is not clear from the outset of the paper (including in the abstract) that the protocol for EWS review is specific to adult inpatients in hospital and this would be good to clarify from the beginning.

In relation to the review selection criteria it might be useful to consider structuring it according to PICOS or referring to these.

Response: This is a very good point. We have clarified at various points in the manuscript that we are planning to investigate early warning scores for adult hospital inpatients.

VERSION 2 – REVIEW

REVIEWER	Veronica Lambert Dublin City University No Competing Interest
REVIEW RETURNED	01-Oct-2017
GENERAL COMMENTS	Has made some edits based on previous review feedback